# Tribological Performance of Nanocomposite Carbon Lubricant Additive

**DOI:** 10.3390/ma12010149

**Published:** 2019-01-04

**Authors:** Chuanyi Xue, Shouren Wang, Daosheng Wen, Gaoqi Wang, Yong Wang

**Affiliations:** 1School of Mechanical Engineering, University of Jinan, Jinan 250022, China; jdcyjj@163.com (C.X.); me_wends@ujn.edu.cn (D.W.); me_wanggq@ujn.edu.cn (G.W.); 2School of Physics and Technology, University of Jinan, Jinan 250022, China; ss_wangy@ujn.edu.cn

**Keywords:** nanocomposite carbon, lubricant additive, tribological properties, anti-wear

## Abstract

In this research, nanocomposite carbon has been found to have excellent tribological properties as a lubricant additive. To reduce high friction and wear in friction pairs, the modified nanocomposite carbon has been prepared for chemical technology. The morphology and microstructure of the modified nanocomposite carbon were investigated via TEM, SEM, EDS, XPS, and Raman. In this study, varying concentrations (1, 3, and 5 wt. %) within the modified nanocomposite carbon were dispersed at 350 SN lubricant for base oil. The suspension stability of lubricating oils with the modified nanocomposite carbon was determined by ultraviolet-visible light (UV-VIS) spectrophotometry. The friction and wear characteristics of lubricants containing materials of the modified nanocomposite carbon were evaluated under reciprocating test conditions to simulate contact. The morphology and microstructure of the friction pair tribofilms produced during frictional contact were investigated via SEM, EDS, and a 3D surface profiler. The results showed that scratches, pits, grooves, and adhesive wear were significantly reduced on the surface of the friction pair which was used with 3% nanocomposite carbon lubricant. Additionally, the modified nanocomposite carbon showed excellent friction reducing and anti-wear performance, with great potential for the application of anti-wear.

## 1. Introduction

With the advancement of science, the call for energy conservation and emission reduction has been very high. Everywhere in the world, energy conservation has become more and more popular. In terms of energy loss, friction losses account for 40% of energy losses. Engine loss accounts for 60% of vehicle losses [1]. Reducing engine wear has become a top priority. Additionally, now, there are many problems with engine oil: Extreme pressure, wear resistance, corrosive metal, and a polluted environment. In order to solve these problems, it has become popular to add carbon nano-additives to lubricating oils to improve the quality of lubricating oils. Most of this solid lubricant forms a lubricating film on the surface of the metal, thereby preventing the direct contact of the metal to exacerbate wear [2]. The general lubrication state is divided into boundary lubrication, mixed lubrication, and hydrodynamic lubrication. In work, the friction pair generally passes through these lubrication states. The order of film thickness from small to large is boundary lubrication, mixed lubrication, and hydrodynamic lubrication. Experimental studies have found that nanocomposite carbon lubricant additives exhibit superior antifriction and anti-wear properties. The main component of nanocomposite carbon additive is ultra-dispersed nanodiamond-graphite powder, which was first prepared by detonation in 1980 [3]. Nanocomposite carbon has the advantages of an extremely small size, high adsorption, high hardness, large specific surface area, and high surface activity. Because of its excellent performance, it is very suitable for lubricant additives. Domestic and foreign experts are watching Nanocomposite carbon. Nanocomposite carbon is used in lubricating oil. Additionally, nanocomposite carbon is agglomerated by the influence of van der Waals force and electrostatic force [4].

Many experts have conducted research on the lubrication of nanocomposite carbon. Some efforts have been made to solve the problem of nanocomposite carbon surface modification technology and agglomeration. Neverovskaya A Yu et al. [5] designed a chemical modification technique for the surface of nanodiamonds by silane-based grafting, and proposed a hydrogen-bonded stable polymerization model formed by different types of functional groups. Suzuki et al. [6] found that ultrasound can reduce the agglomeration of ultra-dispersed diamond in lubricating oil. Vadym et al. [7] discussed the rational control of the chemical, electronic, and optical properties of nanodiamonds by surface doping and functional group introduction. After a reasonable analysis of the surface modification in principle, the friction and wear test was designed. Ming et al. [8] used relative optical interference (Roll) technology and a high-precision force measurement system to detect the oil film thickness and friction of the ultra-dispersed nanodiamond lubricant. They found that the oil film containing the ultra-dispersed nanodiamond lubricant was thicker and had many grooves on the surface to reduce friction. It has been found that nanocomposite carbon has excellent tribological properties through friction and wear tests. Zhang et al. [9] found that ultra-dispersed nanodiamonds exhibit excellent load-carrying capacity and anti-wear properties in the formation of liquid-solid two-phase systems between friction pairs and lubricating oils. Chu et al. [10] studied the friction properties of nanodiamonds added to lubricating oils and found that they can delay or even avoid scratches on the surface of the friction pair, showing good performance. Wang et al. [11] dispersed nano-copper, nano-iron, and nanodiamond particles into semi-finished lubricants. The nanodiamond lubricant film has very good anti-sliding performance and load-bearing capacity, and has a repair function for the friction pair. Nanocomposite carbon has been studied by researchers for different friction pair materials. C-C chou et al. [12] studied the rheological properties of ultra-dispersed nanodiamonds and the tribological properties of two carbon steels and aluminum alloys. It was found that nanodiamonds have greatly improved the anti-wear ability to carbon steel. Nanodiamond dispersion improves the surface roughness of carbon steel. The wear-reducing ability of an aluminum alloy mainly depends on the viscosity of the nanodiamond dispersion. Hwang et al. [13] studied the lubricant containing nanodiamond particles and found that the tribological properties of S45C carbon steel can be improved. The best anti-wear effect is 0.0075%. Tribological tests of various contact types of nanocomposite carbon were summarized. H Huang et al. [14] studied the tribological properties of nanodiamond lubricants in the form of point contact, line contact, and surface contact, and determined the optimal addition amount of nanoparticles under various contact conditions. Different forms of friction pairs exhibit different tribological properties. In order to combine the tribological test of nanocomposite carbon with the actual working conditions, the bench test was performed. RedKin VE [15] found that the nanodiamond-graphite composite additive was added to the engine oil, and the anti-wear performance of the oil was significantly improved by 20–30%, and the noise was reduced. Zhang W et al. [16] found the superiority of nanodiamonds under extreme pressure and anti-wear. Engine bench tests show that the power, economy, emissions, detergency, and sediments were significantly improved, and steel wear can be reduced by 82%. The oil film bearing capacity was greatly improved, and the frictional power consumption and oil temperature were significantly reduced. The transmission mechanical efficiency was improved, and the friction pair thus has a self-repairing ability.

Summarizing the research of nanodiamond-graphite lubricating oil, authors have found that some problems, such as the dispersibility, stability, uniformity, and medium compatibility, of nanodiamond-graphite in lubricating oil have not been solved. In this paper, the surface modification of nanocomposite carbon is carried out by using a reagent, such as a titanate coupling agent. Nanocomposite carbon lubricant additives and lubricating oils are prepared to use a dispersant to maintain stability. According to the standard design of reciprocating friction and wear tests under different operating conditions, and different contact loads and nanocomposite carbon concentrations were selected. In addition, field emission electron microscopy (FE-SEM), transmission electron microscope (TEM), Energy Dispersive Spectrometer (EDS), Raman spectroscopy, X-ray photoelectron spectroscopy (XPS), and 3D profiler were used to analyze nanocomposite carbon and wear. The main mechanism of improving the friction and wear of nanocomposite carbon lubricant in reciprocating friction and wear experiments was explored.

## 2. Materials and Methods 

### 2.1. Materials

Nanocomposite carbon were provided by Fu Jian Shishi Materials Tech Co., Ltd (Quanzhou, China). The average sizes of the nanocomposite carbon were 5 nm. T-161A high molecular weight polyisobutylene succinimide, T-109 calcium salicylate, and alcohol were all obtained from Xilong Chemical Co., Ltd (Shantou, China). and the 350 SN lubricant was supplied by the Tianshi Lubricating Oil Co. (Suzhou, China). The weight percent of the modified nanocomposite carbon additives in base oil were 1, 3, and 5 wt. %. The tribological tests were conducted by using base oil (350 SN) to demonstrate the effect of the modified nanocomposite carbon as engine oil additives. Other material parameters are shown in Table 1.

### 2.2. Nanocomposite Carbon Lubricating Oil Additive Fabrication

The surface of nanocomposite carbon adsorbs many hydrophilic groups, such as hydroxyl groups and carboxyl groups. To stabilize it in the lubricating oil, surface modification treatment and dispersant are needed to improve the stability of nanocomposite carbon in the lubricating oil. The ball milling modification method and the dispersing agent were used together. In order to maximize the effect of the titanate coupling agent, it was necessary to add a diluent of 350 SN and a titanate coupling agent 1:1. The stainless steel ball was used as a ball milling medium. The ball to material ratio was 3:1. nanocomposite carbon and the diluted titanate coupling agent were ball milled in a ball mill for 5 h to obtain surface modified nanocomposite carbon [17]. Then, nanocomposite carbon additive dispersion solution was prepared by mixing T161A high molecular weight polyisobutenyl succinimide and 350 SN lubricants in a 1:1 ratio. The preparation of the nanocomposite carbon additive was a mixture of the modified nanocomposite carbon and the dispersion solution ratio of 1:39. The mixed solution was transferred to a magnetic stirrer and heated to a temperature of 80 °C for 1 h with stirring. Finally, the solution was washed to neutral by detergent T-109 to obtain the nanocomposite carbon additive. Nanocomposite carbon lubricants are made by adding different concentrations of nanocomposite carbon additives to the 350 SN base oil. The mixed solution was ultrasonically dispersed for 20 min. Different concentrations of nanocomposite carbon lubricants were obtained.

### 2.3. Characterizations of Nanocomposite Carbon

X-ray photoelectron spectroscopy measurements were carried out with an AXIS Supra by Kratos Analytical Inc. (San Diego, CA, USA). using monochromatized Al Kα radiation (hv = 1486.6 eV, 150 W) as the X-ray source with a base pressure of 10^−9^ torr. Survey scan spectra were acquired using a pass energy of 160 eV and a 1 eV step size. Narrow region scans were acquired using a pass energy of 40 eV and a 0.1 eV step size. The hybrid lens mode was used in both cases. The analyzed area of all XPS spectra was 300 μm × 700 μm. Transmission electron microscope (TEM) (H-8100, HITACHI, Tokyo, Japan) was used to study the microscopic morphology of nanocomposite carbon. The lattice resolution and dot resolution of the transmission electron microscope were 0.144 nm and 0.205 nm. The scanning transmission function resolution was 1.5 nm. TEM sample preparation was ultrasonically shaken for 10 to 30 min using an appropriate amount of nanocomposite carbon and ethanol. The obtained uniform mixture was sucked out with a glass capillary, and then 2 to 3 drops of the mixed liquid were dropped onto the microgrid. Scanning electron microscopy (SEM) (SUPRA^TM^ 55, Zeiss, Jena, Germany was adopted to investigate the morphology of the prepared materials. Ultraviolet-visible spectrophotometry (UV-VIS) (CARY 300) (Agilent, Palo Alto, CA, USA) was often used to detect nanocomposite carbon and the modified nanocomposite carbon concentration. Energy Dispersive Spectrometer (EDS) (Oxford, Oxford, UK was used for surface composition analysis. Raman spectroscopy is an analytical method used in molecular structure research. The surface characteristics of the wear material were characterized by 3D microscope with a super wide depth of field (VHX-700F,) ( KEYENCE, Tokyo, Japan).

### 2.4. Tribological Testing

The solvent neutral (350 SN) mineral oil was used as a lube base oil for the tribo-evaluation of the modified nanocomposite carbon. The dispersions of the variable mass ratio (1, 3, and 5 wt. %) of the modified nanocomposite carbon in the 350 SN base oil were prepared by using an ultrasonic bath. The tribological properties of the modified nanocomposite carbon, which was under EP conditions, were detected with a reciprocating tribotester (UMT) (BRUKER, Karlsruhe, Germany (Figure 1), following the ASTM 181-11 test method for testing lubricants under scuffing conditions [18]. In Figure 1, the modified nanocomposite carbon is in contact with each other on the metal surface. As the concentration of the reciprocating nanocomposite carbon molecules fill the pits on the metal surface, it is uniformly dispersed, like nanocomposite carbon in the model diagram. The friction experiments were carried out on a reciprocating sliding tester with a load of 50, 100, 250, and 400 N. A sliding stroke and a frequency of 1 mm and 10 Hz were chosen for obtaining the boundary lubrication condition. The tribological behavior of the modified nanocomposite carbon lubricants were evaluated by using a reciprocating tribometer, which was under operating conditions to the disk-to-disk interface. The coefficient of friction was recorded as a function of time, and the wearing capacity was computed based on multiple measurements of each sample. In each tribo-test, the upper and the lower specimens were ultrasonically cleaned in acetone for 15 min, which could remove any dust and grease on surfaces. The surface was polished before each test. The friction surface wear test was performed after the surface had reached the roughness standard (0.43 and 0.45 microns). The results of wear during this investigation are presented in terms of wear, which were calculated using weighing [19]. 

## 3. Results

### 3.1. Structure Characteristics of As-Prepared Nanocomposite Carbon Additives

Figure 2 shows the TEM micrograph with the morphology of the nanocomposite carbon used for this study. The nanodiamond-graphite powders in nanocomposite carbon were prepared by detonation. The nanodiamond-graphite in nanocomposite carbon had an average particle size of only 5 nm. It can be seen in Figure 2 that this nanocomposite carbon had an ordered diamond core.

Figure 3 shows SEM images of the modified nanocomposite carbon. By comparing these two images, it is clear that the microstructure of the modified nanocomposite carbon was greatly different in the unmodified state. In Figure 3b, the ×10,000 magnified image of the modified nanocomposite carbon shows the schistose structure. Additionally, the ×100,000 magnified image of the modified nanocomposite carbon shows the flocculent structure in Figure 3d. Further, the modified nanocomposite carbon was analyzed and studied.

The EDS patterns with elemental content of the modified nanocomposite carbon are shown in Figure 4. The modified nanocomposite carbon morphology may be affected by organic, but it is not sure about the specific changes by EDS analysis. The EDS element mapping in Figure 4 displays the chemical composition of the modified nanocomposite carbon, and which new element was found in addition to carbon, such as sulfur, calcium, zinc, and so on. The emergence of new elements was mostly due to the residual of the modified reagents. Additionally, nanocomposite carbon sheet was attached to the chemical effect of the modified solution, in which the nanocomposite carbon structure can be covered by the original [20]. Consequently, the microstructure of nanocomposite carbon underwent great changes because of the influence of the modification treatment. Nanocomposite carbon was used in this state to help stabilize the suspension in lubricating oil.

Raman spectroscopy is a means of studying carbon structure efficiently and can characterize the morphology of carbon. The Raman spectrum of the nanocomposite carbon additive is shown in Figure 5. As can be seen from Figure 5, the nanodiamond surface of nanocomposite carbon should be covered with graphite. The broad Raman peak near 1329 cm^−1^ is a characteristic peak of the sp^3^ structure nanodiamond, and the G-band observed at around 1580 cm^−1^ of the Raman peak is a nanographite characterizing the sp^2^ structure [21]. The peak position of the surface modified Raman spectrum did not change, but the absorption intensity decreased. Since the Raman scattering cross section of diamond is 1/60 of graphite, this indicates that nanocomposite carbon has both nanodiamond and sp^2^ structure nanographite residues [22].

Figure 6 shows the XPS spectrum of nanocomposite carbon. Raman spectroscopy is a qualitative analysis. However, XPS is used for quantitative analysis. XPS is used to excite a solid surface with a beam of X-rays while measuring the kinetic energy emitted from the surface of the material being analyzed at 1–10 nm to obtain an XPS spectrum [23]. Electrons exceeding a certain kinetic energy are recorded by photoelectron spectroscopy. The peak appearing in the photoelectron spectrum is the emission of a certain characteristic energy electron in the atom. Each element has a specific core electron binding energy. The energy and intensity of the photoelectron peaks are used to qualitatively and quantitatively analyze all elements [24]. Figure 6a shows the total elemental peak of nanocomposite carbon. Nanocomposite carbon was quantitatively analyzed by the narrow peaks of all the elements to obtain a C content of 92.97%, an O content of 4.56%, and an N content of 1.96%. Comparing with the elements obtained from the wide peaks, it was found that there was an error in the wide peak. The presence of the N element on the surface of nanocomposite carbon was applied. It is the most widely used impurity element in nanocomposite carbon [25]. In order to analyze the content of different species of C, the C1s spectrum was subjected to a calibration analysis using the Shirley method to subtract background peaks. Figure 6b shows a partial-peak fit of the C1s spectrum using a Gaussian-Lorentzian fit. The content of carbon is represented by the characteristic peak area obtained by fitting. C-sp^3^ is a diamond characteristic peak with a position of 284.79 eV and an area ratio of 60.01%. C-sp^2^ is a characteristic peak of graphite with a position of 283.49 eV and an area ratio of 26.87%. The remaining characteristic peaks are C-O (286.18 eV) and C=O (288.01 eV). The main forms of nanocomposite carbon were found to be diamond and graphite by XPS [26].

Figure 7 shows that the suspension stability of the 5% modified nanocomposite carbon and nanocomposite carbon lubricants was detected by UV spectrophotometer. On the one hand, the dispersion stability of the modified and unmodified nanocomposite carbon to the 350 SN base oil was considered. On the other hand, the suspension stability was affected by the excessive concentration [27]. Therefore, the tested concentrations of the modified and unmodified nanocomposite carbons were determined to be 5%. The particle concentrations of the two different suspensions were nearly equal before centrifugation. After centrifugation, the unmodified nanocomposite carbon suspension was allowed to stand for 20 min, and the precipitation of unmodified nanocomposite carbon was observed. It indicates that the unmodified nanocomposite carbon appeared as an agglomeration in the base oil. In contrast, a small amount of precipitate was observed in the modified nanocomposite carbon suspension. The results show that the modified nanocomposite carbon has an excellent stable suspension. Additionally, this improvement can be attributed to the effectiveness of the surface modification. After the modification test, the hydroxyl group on the surface of nanocomposite carbon reacts with the hydroxyl group of the titanate coupling agent to make nanocomposite carbon exhibit lipophilicity. The affinity between the organic monomer and the inorganic powder was improved [28]. When the modified nanocomposite carbon was dispersed in the base oil, the long chain hydrocarbons easily penetrated the base oil. So, the modified nanocomposite carbon appeared with a typical hindrance steric effect, which can be separated from each nanocomposite carbon molecule [29]. At the same time, the steric hindrance force overcame the gravity and prevented the aggregation of the nanocomposite carbon sheet. Therefore, the 5% modified nanocomposite carbon lubricants formed a uniform and stable suspension.

### 3.2. Effect of Modified Nanocomposite Carbon Content on the Friction and Wear Properties of Friction Pairs

Figure 8 shows the friction coefficient curve and the wear percentage loss of the friction pair of the 350 SN base oil at different concentrations of the modified nanocomposite carbon at an average sliding speed of 0.4 m/s and a contact load of 250 N. The coefficient of friction was reduced by the addition of the modified nanocomposite carbon. As the concentration of the modified nanocomposite carbon increased, the friction coefficient decreased first and then increased. Figure 8a shows that the best result was 3% modified nanocomposite carbon. The particles exhibited anti-friction properties, and the modified nanocomposite carbon can be used as a ball of the friction pairs or form a continuous lubricating film to reduce the friction coefficient [30]. The micro-pits the friction pairs were filled with nanocomposite carbon to increase the actual contact area of the friction pairs and reduce the surface roughness. As can be seen from Figure 8b, the mass loss of the friction pair was reduced by the addition of the modified nanocomposite carbon. As the concentration of the modified nanocomposite carbon increased, the mass loss of the friction pair first decreased and then increased. In particular, the concentration of the modified nanocomposite carbon was higher than 3%, and the mass loss of the friction pair was increased.

Figure 9 shows the average friction coefficient and wear loss percentage of 350 SN engine oils with varying concentrations of the modified nanocomposite carbon under a 0.4 m/s average sliding speed and a contact load of 50, 100, 250, and 400 N. The curves in Figure 9a exhibit the average friction coefficient significantly reducing under high loads (250 N, 400 N) and a lower improvement found at low loads (50 N, 100 N). At a load of 250 and 400 N, the modified nanocomposite carbon was the best concentration of 3%. The modified nanocomposite carbon particles exhibited friction reduction under high load, and it was possible to form a lubricating film on the modified nanocomposite carbon particles and the friction side table due to high load conditions, thereby lowering the friction coefficient [31]. With the increase of the load, the wear loss of the friction pair also increased, as shown in Figure 9b. Specifically, at loads of 50 and 100 N, the friction pair wore less. When the load exceeded 250 N, the amount of wear increased from the load. The load reached 400 N, and the wear was more serious. Under different loads, the modified nanocomposite carbon concentration above 3% resulted in increased friction pair wear. However, this difference is that the friction pair wore more severely at high loads.

The 40 °C and 100 °C kinematic viscosities of 350 SN base oil and 350 SN lubricant with 3% and 5% nanocomposite carbon are presented in Figure 10a. The kinematic viscosity of the 350 SN base oil and the added nanocomposite carbon lubricant were tested using the lubricant test standards, GB/T 265-1988 and GB/T 1995–1998 [32,33]. By increasing the temperature, the kinematic viscosity of the 350 SN base oil and nanocomposite carbon lubricant added decreased. According to Figure 10a, it was found that the 350 SN lubricant with 3% and 5% of nanocomposite carbon added had a slight decrease in kinematic viscosity at 40 °C and 100 °C. The change in kinematic viscosity of the current nanocomposite carbon lubricant can be regarded as constant [34].

The Stribeck curve obtained by the friction test is shown in Figure 10b. This test evaluated the tribological properties of nanocomposite carbon lubricants under different lubrication conditions. The kinematic viscosity are shown in Figure 10a. The average sliding speed was constant. The Stribeck curve only related to the effects of the load. This method can determine the specific lubrication mode and the range of coefficients of friction. The high load was maintained to maintain the boundary lubrication state. The Stribeck parameter was defined here as [35]:(1)$$stribeck\;parameter=\frac{dynamic\;viscosity\times sliding\;velocity}{load\;unit}$$

It can be seen that the use of lubricants reduced the coefficient of friction by 47–50%, 30–44%, and 8–13%, under boundary, mixing, and fluid dynamic lubrication. Point B means that the friction coefficient changed from hybrid lubrication to approximately hydrodynamic lubrication due to forming a solid protective layer with low shear stress. The positive effect of the transition B occured at lower Stribeck parameters and resulted in reduced wear during boundary and mixed lubrication conditions. 

### 3.3. Worn Surface Analyses 

Figure 11 shows the SEM images of the worn surface lubricated by 350 SN lubricant under the condition of 50 N–10 Hz. The wear track for the base lubricant presented many grooves and furrows, as well as some metal burr due to adhesive wear in the absence of a tribofilm. The surface of the friction pair appeared as pits and furrows. One part of the surface was smooth, the other part of the surface was damaged. The main reason may be that the surface of the friction pair can form a discontinuous lubricating film [36]. Additionally, the generating abrasive particles were increased in terms of the wear.

Figure 12 shows the microscopic morphology of the surface abraded with the modified nanocomposite carbon lubricant. By comparing the surface before sliding, there were very obvious changes to the worn surfaces. The worn surfaces did not appear with a large number of pits and furrows, and it was relatively smooth. So, the modified nanocomposite carbon lubricant thus reduced fatigue loss. The reason for this is that the nanocomposite carbon has a strong adhesion capacity on the metal surface, which is attached to forming a layer of nanocomposite carbon film [37].

The 3D morphology of the wear material was characterized by 3D microscope with a super wide depth of field. The 3D morphology of the wear surface of different mass fraction nanocomposite carbon additive lubricants is shown in Figure 13. The distance between the peak and valley of the surface profile was denoted by DPV [38]. The smallest DPV in Figure 13a is 7.6 μm. With the increasing concentration, the wear scar depth was significantly increased. Also, many pits were found on the surface after the test. Soon after, the DPV gradually increased. The maximum DPV of the sample at a concentration of 5% was 15.09 μm (Figure 13d), 0% was 8.4 μm (Figure 13c), and 1% was 7.88 μm.

The surface roughness of the friction pair is shown in Figure 14. Figure 14a shows that the friction pair with a 3% nanocomposite carbon additive had a surface roughness of at least Ra of 0.18 μm (Ra is the arithmetic mean deviation of the contour). The surface roughness of Figure 14b is slightly higher than Ra of 0.24 μm. This phenomenon may be due to the surface micropolishing ability of nanocomposite carbon to smooth the surface irregularities. The concentration of nanocomposite carbon additive was too large, and the surface roughness was increased to Ra of 0.54 μm. It was higher than Ra of 0.44 μm without the addition of nanocomposite carbon. Sia et al. obtained similar results [39].

## 4. Conclusions

The microstructure of nanocomposite carbon before the modification was a cell shape, while the microstructure of the modified nanocomposite carbon was flocculent. The group, such as the hydroxyl group, on the surface of nanocomposite carbon reacted with the titanate coupling agent to mix the dispersant and the cleaning agent, and was finally stably suspended in the lubricating oil. Through Raman spectroscopy and XPS analysis, it was found that nanocomposite carbon is mainly composed of nanodiamond and nanographite. No substance change occurred before and after the modification, but only impurity elements, such as sulfur, calcium, and the like, were introduced.Under low load (50 N, 100 N), the modified nanocomposite carbon particles had little effect on the friction coefficient of the friction pair; under high loads (250 N, 400 N), the modified nanocomposite carbon particles can reduce the friction coefficient of the friction pair. Among them, when the mass fraction of the modified nanocomposite carbon in 350 SN lubricant was 3%, the friction reduction effect was the best. Nanocomposite carbon lubricants exhibited good surface micropolishing.The antifriction mechanism of the modified nanocomposite carbon particles in the friction pair: The modified Nanocomposite Carbon particles participated in the formation of the lubricating film under a high load. Under the action of friction shearing and frictional heat, the modified nanocomposite carbon particles formed a lubricating film on the surface of the friction pair; under a low load, the modified nanocomposite carbon particles only deposited and adsorbed on the surface of the friction pair. The deposited modified nanocomposite carbon did not participate in the formation of a lubricating film, so it did not exhibit a friction reducing effect.

## Figures and Tables

**Figure 1 materials-12-00149-f001:**
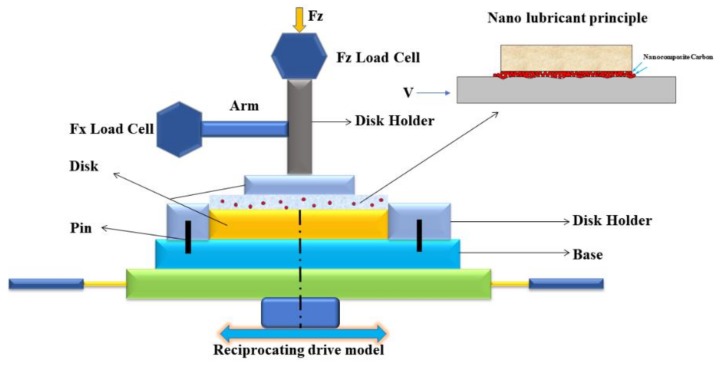
Photograph of sample holders in the UMT test chamber.

**Figure 2 materials-12-00149-f002:**
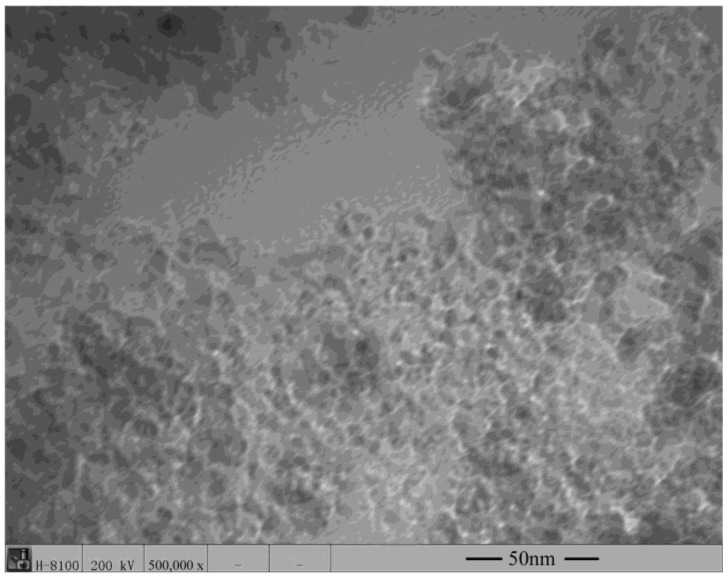
TEM images of nanocomposite carbon.

**Figure 3 materials-12-00149-f003:**
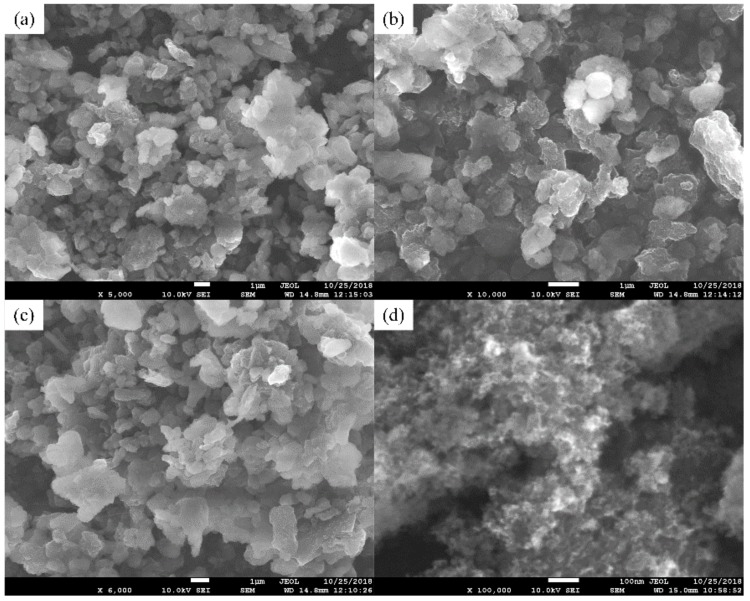
SEM images of the modified nanocomposite carbon: (**a**) ×5000, (**b**) ×10,000, (**c**) ×6000 WD = 14.8 mm, and (**d**) ×100,000 WD = 15.0 mm.

**Figure 4 materials-12-00149-f004:**
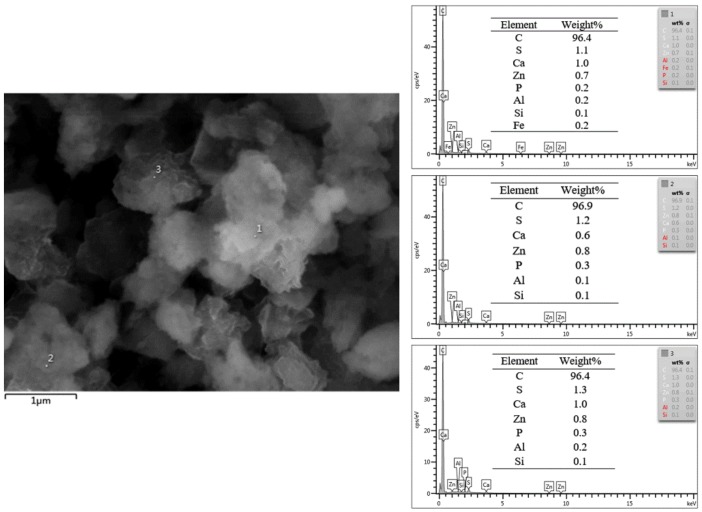
EDS patterns of the modified nanocomposite carbon surface with different positions.

**Figure 5 materials-12-00149-f005:**
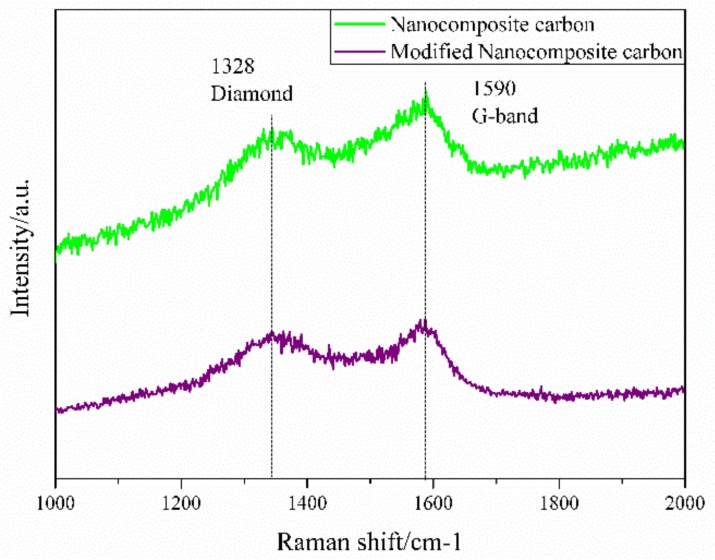
Raman spectra of nanocomposite carbon before and after modification.

**Figure 6 materials-12-00149-f006:**
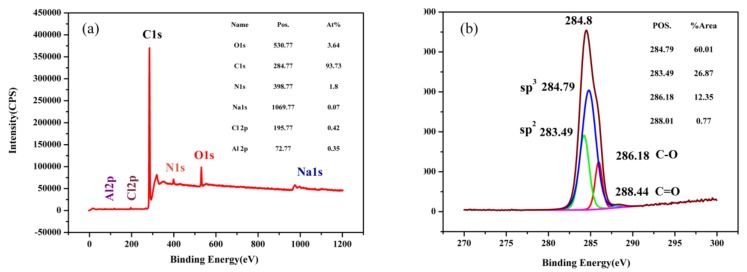
XPS spectrum of nanocomposite carbon: (**a**) Wide spectrum; (**b**) C1s spectrum.

**Figure 7 materials-12-00149-f007:**
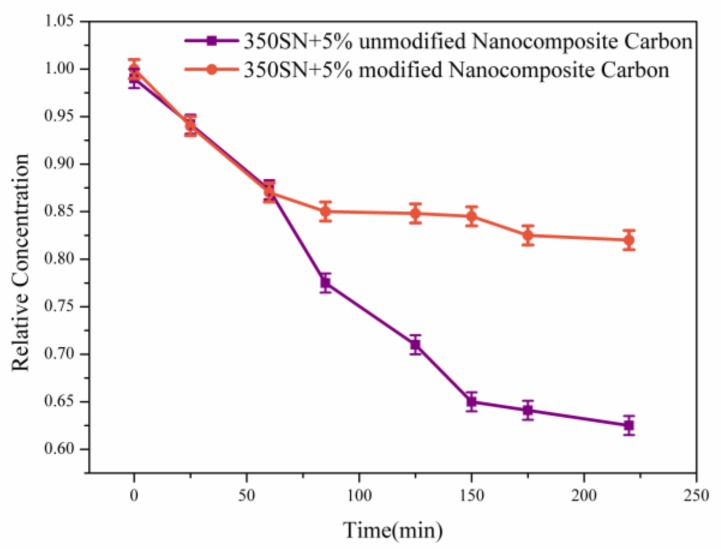
Suspension stability of the lubricating oils with 5% modified nanocomposite carbon and nanocomposite carbon as determined by ultraviolet-visible light (UV-VIS) spectrophotometry.

**Figure 8 materials-12-00149-f008:**
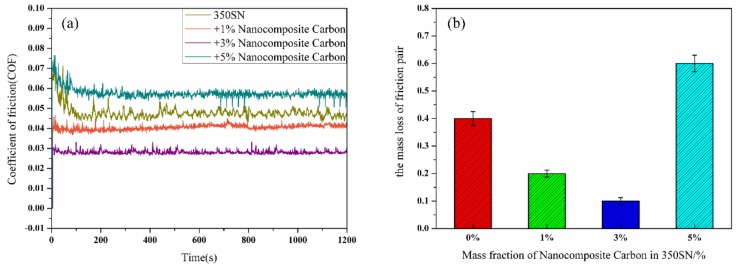
(**a**) Coefficient of friction; (**b**) the friction pairs’ mass loss.

**Figure 9 materials-12-00149-f009:**
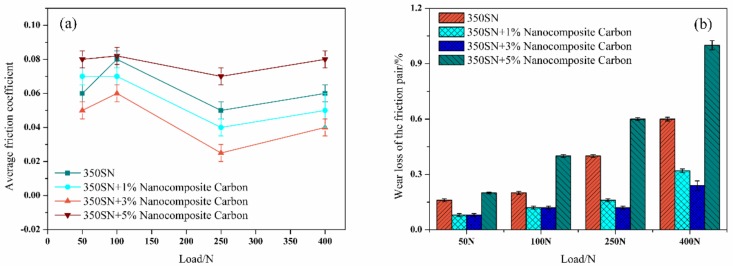
(**a**) Average friction coefficient; (**b**) wear loss of the friction pair.

**Figure 10 materials-12-00149-f010:**
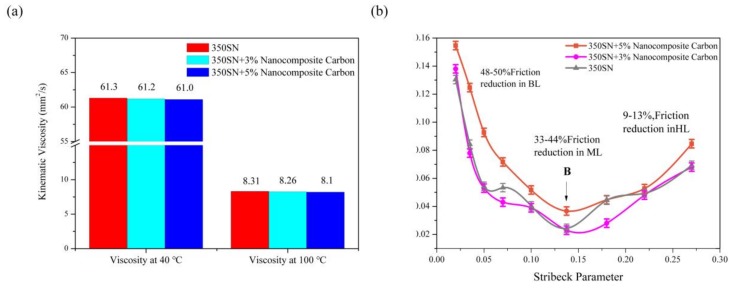
350 SN with and without the use of nanocomposite carbon additives: (**a**) Kinematic viscosity; (**b**) experimental Stribeck curve for assembly.

**Figure 11 materials-12-00149-f011:**
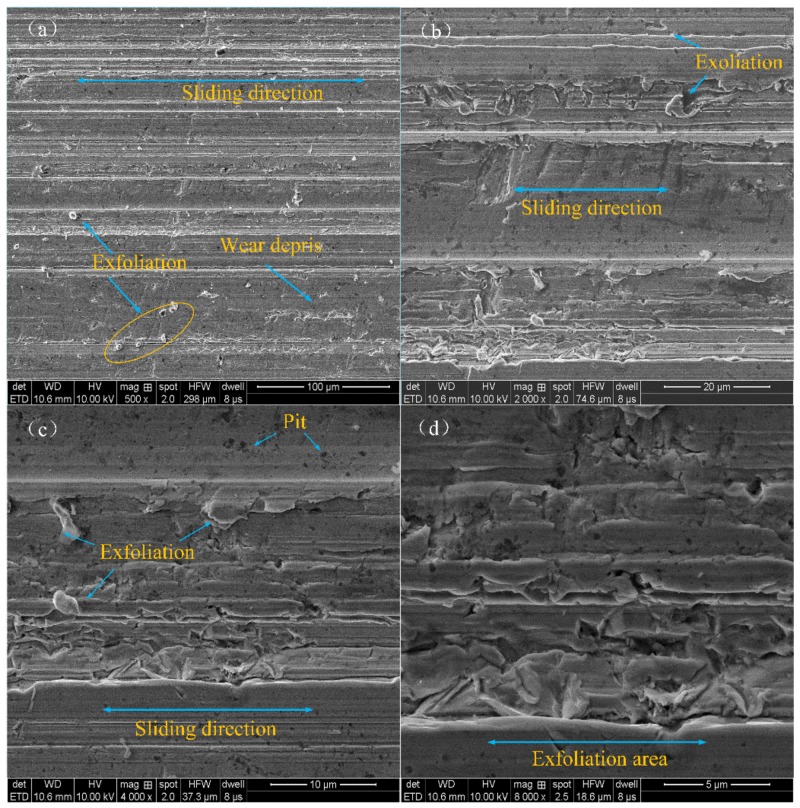
Typical SEM images of wear scars lubricated with 350 SN: (**a**) ×500, (**b**) ×2,000, (**c**) ×4,000 and (**d**) ×8,000.

**Figure 12 materials-12-00149-f012:**
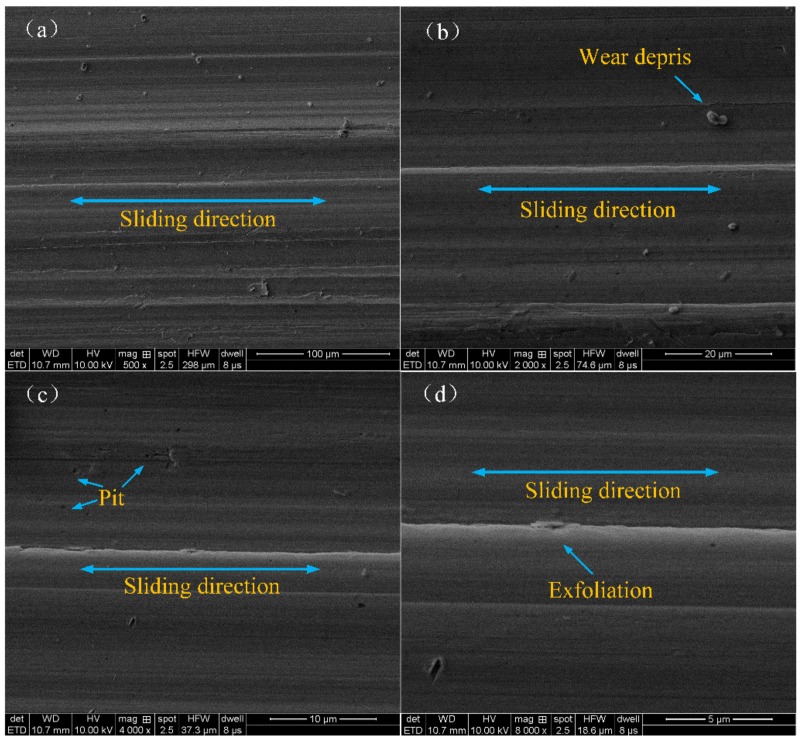
SEM images of wear scars lubricated with 3% modified nanocomposite carbon: (**a**) ×500, (**b**) ×2,000, (**c**) ×4,000 and (**d**) ×8,000.

**Figure 13 materials-12-00149-f013:**
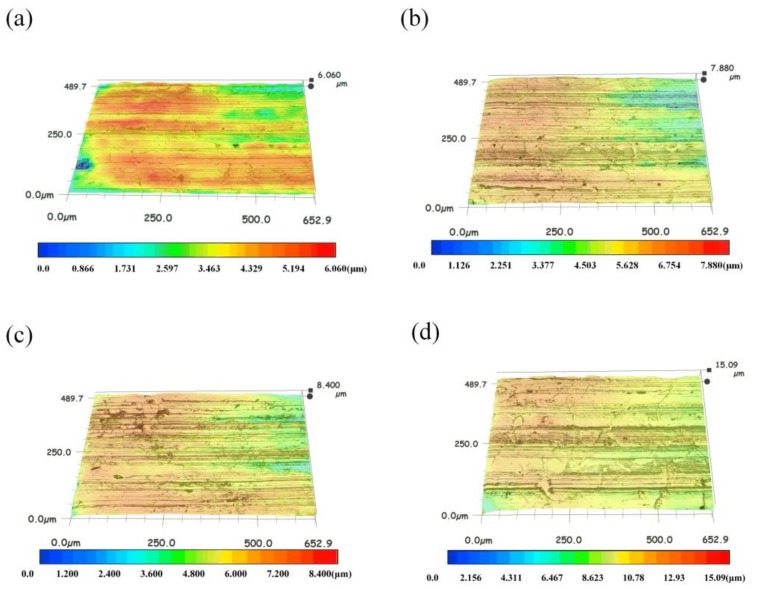
3D morphologies of the friction pair worn surfaces for (**a**) 3%, 6.06 μm, (**b**) 1%, 7.88 μm, (**c**) 0%, 8.40 μm, and (**d**) 5%,15.09 μm.

**Figure 14 materials-12-00149-f014:**
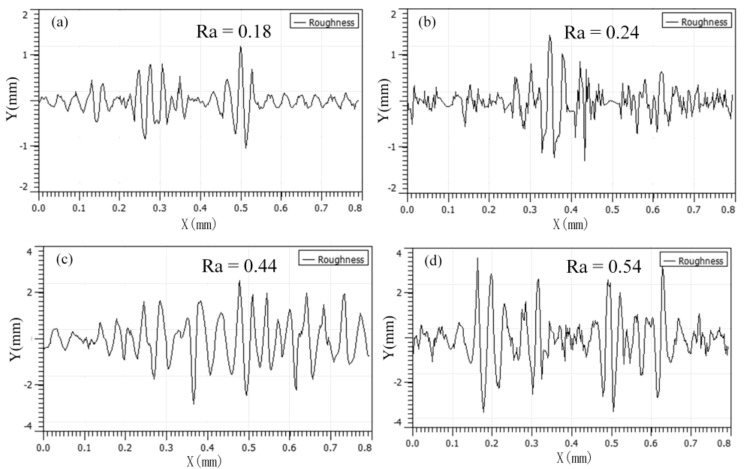
Surface roughness profiles for blocks lubricated with: (**a**) 3%, (**b**) 1%, (**c**) 0%, and (**d**) 5%.

**Table 1 materials-12-00149-t001:** Material properties.

Material	Properties
**Specimens**	
Up	AISI D2 steel, dimensions: 6 × 17 × 10 mm, Ra = 0.43 μm, hardness: 62 HRC
Down	AISI 1018 steel, dimensions: 45 × 34 × 8 mm, Ra = 0.45 μm, hardness: 78 HRB
**Testing conditions**	
Reciprocating frequency	10 Hz
Stroke	30 mm

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
