# Peer review of "Tribological Performance of Nanocomposite Carbon Lubricant Additive"

_materials, 2019, doi:10.3390/ma12010149_

Reviewer 1 Report

The authors have written an interesting manuscript in which they analyze the usage of Nanocomposite Carbon materials as additive in lubrication. The paper is well structured, with firstly a characterization of the nanomaterial and afterwards the determination of the tribological performance of the lubricants, using three concentration of additive. The research is good, but there are some points to be improved for publication:

Major points

The manuscript redaction does not achieve the standards of a journal like Materials. There are very many typos and I recommend a profound revision of the English grammar and the style in general. For example:

Line 43 “anocomposite”

Line 52 and 53, in this sentence the authors refer to the same article than line 50.

Line 63 “It was found that nanodiamonds have greatly improved the ability to reduce carbon steel”. This sentence is very unclear.

Line 68, “disperse” instead of “dispersed”.

Line 204, “The content of each element in the nanocomposite carbon is quantitatively analyzed”. If this sentence is removed, we do not lose any kind of information.

In general, the style of the text from line 49 to 82 is a description of articles without cohesion.

Line 280 “parameteris”

Line 286 “Stribecle”

Regarding the scientific aspects, in Figure 10 the authors have represented the Stribeck curves. This parameter is calculated with the values of equation (1). I wonder about the viscosity values. First of all, I would like to see these values in a table, and how viscosity changes with the concentration of nanomaterial. Maybe, an interesting discussion can be included. However, my main concern is that they may have a problem with the behavior of the lubricant. In this article “International Communications in Heat and Mass Transfer, 76, 308–315 (2016)”, a Newtonian behavior is found, and the viscosity can be considered constant.  In contrast, in this other article “Tribology International, 122, 200-209 (2018)” the authors describe a Non-Newtonian behavior of a carbon-based nanofluid, therefore, the viscosity changes with the values of the sliding velocity (which can be related to the shear rate of the rheological experiments). This is something to be addressed when the Stribeck parameter is calculated. I also recommend the article "Scaling theory for hydrodynamic lubrication, with application to non-Newtonian lubricants" by Patrick Warren which is published in ArXiv.

Regarding the behavior of the sample with 5 % of nanomaterial, the SEM and RAMAN analysis of the debris particles are needed. At this high concentration the aggregation of the particles may occur more likely, and these aggregates can increase friction and wear. Size and composition can be analyzed by SEM and the structure of the carbon nanomaterials by RAMAN of these debris particles. These results can give very valuable information of these systems and the anti-friction mechanism of the lubricant. This is an old discussion which started in 1977 with this work "Wear, 42, 49-62 (1977)" 

Minor points

In section 2.2 the authors describe the fabrication of the nanocomposites, but there are no references at all. I guess that they have based their fabrication method on other articles.

Line 222, the author use the nomenclature “original Nanocomposite”, but I think that it is more correct to say “unmodified” or “pristine Nanocomposite”.

Author Response

Dear Editors and Reviewers:

Thank you for your letter and for the reviewers’ comments concerning our manuscript entitled “Tribological performance of Nanocomposite Carbon lubricant additive” (ID: materials-403185). Those comments are all valuable and very helpful for revising and improving our paper, as well as the important guiding significance to our researches. We have studied comments carefully and have made correction which we hope meet with approval. Revised portion are marked in red in the paper. The main corrections in the paper and the responds to the reviewer’s comments are as flowing:

Point 1: Major points

The manuscript redaction does not achieve the standards of a journal like Materials. There are very many typos and I recommend a profound revision of the English grammar and the style in general. For example:

1.       Line 43 “anocomposite”

Response 1: Line 43, the statements of “anocomposite” were corrected as “Nanocomposite”

2.       Line 52 and 53, in this sentence the authors refer to the same article than line 50.

Response 2: We have re-written this part according to the Reviewer’s suggestion.

3.       Line 63 “It was found that nanodiamonds have greatly improved the ability to reduce carbon steel”. This sentence is very unclear.

Response 3: Line 71-72, the statements of “t was found that nanodiamonds have greatly improved the ability to reduce carbon steel” were corrected as “It was found that nanodiamonds have greatly improved the anti-wear ability to carbon steel.”

4.       Line 68, “disperse” instead of “dispersed”.

Response 4: Line 70, the statements of “disperse” were corrected as “dispersed”

5.       Line 204, “The content of each element in the nanocomposite carbon is quantitatively analyzed”. If this sentence is removed, we do not lose any kind of information.

Response 5: We have removed this line according to the Reviewer’s suggestion.

6.       In general, the style of the text from line 49 to 82 is a description of articles without cohesion.

Response 6: Line 49-89,we have re-written this part according to the Reviewer’s suggestion.

7.       Line 280 “parameteris”

Response 7: Line 306, the statements of “parameteris” were corrected as “parameter is”.

8.       Line 286 “Stribecle”

Response 8: Line 312, the statements of “Stribecle” were corrected as “Stribeck”.

9.       Regarding the scientific aspects, in Figure 10 the authors have represented the Stribeck curves. This parameter is calculated with the values of equation (1). I wonder about the viscosity values. First of all, I would like to see these values in a table, and how viscosity changes with the concentration of nanomaterial. Maybe, an interesting discussion can be included. However, my main concern is that they may have a problem with the behavior of the lubricant. In this article “International Communications in Heat and Mass Transfer, 76, 308–315 (2016)”, a Newtonian behavior is found, and the viscosity can be considered constant.  In contrast, in this other article “Tribology International, 122, 200-209 (2018)” the authors describe a Non-Newtonian behavior of a carbon-based nanofluid, therefore, the viscosity changes with the values of the sliding velocity (which can be related to the shear rate of the rheological experiments). This is something to be addressed when the Stribeck parameter is calculated. I also recommend the article "Scaling theory for hydrodynamic lubrication, with application to non-Newtonian lubricants" by Patrick Warren which is published in ArXiv.

Response 9: Special thanks to you for your good comments. Line 293-300, We have re-written this part according to the Reviewer’s suggestion. We are very sorry for our negligence of Kinematic viscosity. We have added kinematic viscosity change values in Figure 10a. According to Figure 10a, it was found that the kinematic viscosity decreased with increasing concentration, but the change was not large. According to this article “International Communications in Heat and Mass Transfer, 76, 308–315 (2016)”, a Newtonian behavior is found, and the viscosity can be considered constant. This paper also considers the viscosity constant. The sliding speed is also constant. Therefore, the Stribeck curve variable only has a load. 

10.    Regarding the behavior of the sample with 5 % of nanomaterial, the SEM and RAMAN analysis of the debris particles are needed. At this high concentration the aggregation of the particles may occur more likely, and these aggregates can increase friction and wear. Size and composition can be analyzed by SEM and the structure of the carbon nanomaterials by RAMAN of these debris particles. These results can give very valuable information of these systems and the anti-friction mechanism of the lubricant. This is an old discussion which started in 1977 with this work "Wear, 42, 49-62 (1977)"

Response 10: Special thanks to you for your good comments. We are very sorry for our negligence of the behavior of the sample with 5 % of nanomaterial. We are very sorry that we did not express clearly in Figure 7. Figure 7 shows that Suspension stability of the lubricating oils with 5 % modified Nanocomposite Carbon and Nanocomposite Carbon as determined by ultraviolet-visible light (UV-VIS) spectrophotometry. It can be seen that the 5% concentration of nanocomposite carbon lubricant has good dispersion stability. So, we think that the friction and wear are not caused by agglomeration.

 Point 2: Minor points

1.       In section 2.2 the authors describe the fabrication of the nanocomposites, but there are no references at all. I guess that they have based their fabrication method on other articles.

Response 1: Special thanks to you for your good comments. We are very sorry for our negligence of references. We have added references in line 120.

 2.       Line 222, the author use the nomenclature “original Nanocomposite”, but I think that it is more correct to say “unmodified” or “pristine Nanocomposite”.

Response 2: Line 240, the statements of “original” were corrected as “unmodified”.

Thank you very much for consideration!

Sincerely Yours,

Chuanyi Xue

Reviewer 2 Report

First of all a deep English review must be make, as the text is not clearly written. Many typos are also along the manuscript, as example (line 43 a N is missing, 50 use the complete/mixed name of another author, 120 K_alpha,… missing spaces between numbers and units,  lines 324-328 use mm instead of microns…) and many parameters are missed or not well written.

Authors make several experiments using different quantity of additives in a commercial lubricant.

Line 98… they are giving an average size of 4-5 nm… the average is 4? Or 5? Or 4.5? or perhaps sizes range from 4 to 5. The info given in the manuscript has no sense.

In the paper is not clear how authors prepare their additives and the final mix with the lubricant. In line for example 115 they said that they applied a ball-milling to additives but no information about balls used is given.

Line 142. They talk about the pressure applied but they only know the force applied so no the resulting pressure (as pressure is force/area).

In final sections authors compare the surface geometry using different conditions… but in lines 148 they said that the discs used during test were cleaned after each test and used again. What about the surface?

Authors said that they mix the additive and the lubricant with a magnetic stirrer, but conditions indicated in line 114 are not the same as in line 135. What conditions were used?

Line 158 again the average of 4-5 appears again.

In figure 2 is it possible to see that the nanodiamong is encapsulated with graphite??

Paragraph 163-167… what the 10000, 100000 means or correspond to… this is not clear.

Figure 13: why use that order of figures. It will be much clearer for readers to put 0%, 1% 3% and 5%. Same on figure 14.  

Author Response

Dear Editors and Reviewers:

Thank you for your letter and for the reviewers’ comments concerning our manuscript entitled “Tribological performance of Nanocomposite Carbon lubricant additive” (ID: materials-403185). Those comments are all valuable and very helpful for revising and improving our paper, as well as the important guiding significance to our researches. We have studied comments carefully and have made correction which we hope meet with approval. Revised portion are marked in red in the paper. The main corrections in the paper and the responds to the reviewer’s comments are as flowing:

Point 1: First of all a deep English review must be make, as the text is not clearly written.

1.       Many typos are also along the manuscript, as example (line 43 a N is missing, 50 use the complete/mixed name of another author, 120 K_alpha,… missing spaces between numbers and units,  lines 324-328 use mm instead of microns…) and many parameters are missed or not well written. Authors make several experiments using different quantity of additives in a commercial lubricant.

Response 1: Special thanks to you for your good comments. Line 43, 50 and 120, We have re-written this part according to the Reviewer’s suggestion. We have added spaces between units and numbers in the full text. Line 349-355, the statements of “mm” were corrected as “μm”.

2.       Line 98… they are giving an average size of 4-5 nm… the average is 4? Or 5? Or 4.5? or perhaps sizes range from 4 to 5. The info given in the manuscript has no sense.

Response 2: We have made correction according to the Reviewer’s comments. the statements of “size of 4-5 nm” were corrected as “size of 5 nm”.

3.       In the paper is not clear how authors prepare their additives and the final mix with the lubricant. In line for example 115 they said that they applied a ball-milling to additives but no information about balls used is given.

Response 3: Line113-127, We have made correction according to the Reviewer’s comments. We have described the preparation of nanocomposite carbon additives and information on ball milling.

4.       Line 142. They talk about the pressure applied but they only know the force applied so no the resulting pressure (as pressure is force/area).

Response 4: It is really true as Reviewer suggested that the statements of “pressure” were corrected as “load”.

5.       In final sections authors compare the surface geometry using different conditions… but in lines 148 they said that the discs used during test were cleaned after each test and used again. What about the surface?

Response 5: Line 162-163, Dust and grease may remain on the friction side surface during processing. Therefore, we need to clean the friction pair and then carry out the friction and wear test to eliminate the interference.

6.       Authors said that they mix the additive and the lubricant with a magnetic stirrer, but conditions indicated in line 114 are not the same as in line 135. What conditions were used?

Response 6: Line 124-127, Our previous description of lubricant preparation was wrong, and the preparation method of Nanocomposite Carbon lubricating oil was corrected to ultrasonic dispersion.

7.       Line 158 again the average of 4-5 appears again.

Response 7: We are sorry that we have not described clearly. Line 105173the statements of “4-5” were corrected as “5”.

8.       In figure 2 is it possible to see that the nanodiamong is encapsulated with graphite??

Response 8: Line 173We are not rigorously described. the statements of the nanodiamong is encapsulated with graphite were corrected as this Nanocomposite Carbon has a ordered diamond core.

9.       Paragraph 163-167… what the 10000, 100000 means or correspond to… this is not clear.

Response 9: Line 179-181, We have re-written this part according to the Reviewer’s suggestion.

10.    Figure 13: why use that order of figures. It will be much clearer for readers to put 0%, 1% 3% and 5%. Same on figure 14.  

Response 10: Special thanks to you for your good comments. The order used in this paper is to better represent the trend of surface topography. This description is more convenient to understand the trend. So, we describe it this way.

Thank you very much for consideration!

Sincerely Yours,

Chuanyi Xue

Reviewer 3 Report

It is not clearly described, what are the new informations of the work to the other investigations of more groups, you find in part literature.

Many details are failed, specially

row 124 - TEM fabrication, technical data, how do you make TEM sample preparation

row 125 - SupraTM Germany is not the supplierer!

row 162 - in image in row 158-159 the described encapsulation is not to seen

row 165 - 10000 level image .. you will describe 100000 magnification, also row 166

form of the article:

table 1 why doublespace , right column right hand sight 

121 - 10-9    -9 upper case

fig. 3 no mu marker clearly to seen, no information about magnification, working distance

fig. 4 the eds spectra are nothing readable, why energy range over 10 keV nothing to seen

fig 6 the legend is not readable

fig 11 and 12 the text input is to small not readable

how measure the morphology in fig 13. The method is not clearly described AFM?, the x,y,  z-axes levels are not readable

why conclusions in doublespace written?

Author Response

Dear Editors and Reviewers:

Thank you for your letter and for the reviewers’ comments concerning our manuscript entitled “Tribological performance of Nanocomposite Carbon lubricant additive” (ID: materials-403185). Those comments are all valuable and very helpful for revising and improving our paper, as well as the important guiding significance to our researches. We have studied comments carefully and have made correction which we hope meet with approval. Revised portion are marked in red in the paper. The main corrections in the paper and the responds to the reviewer’s comments are as flowing

Point 1: It is not clearly described, what are the new informations of the work to the other investigations of more groups, you find in part literature.

1.       Many details are failed, specially row 124 - TEM fabrication, technical data, how do you make TEM sample preparation

Response 1: We are very sorry for our negligence of many details.

2.       row 125 - SupraTM Germany is not the supplierer!

Response 2: Line 127, the statements of “ SupraTM ” were corrected as “ SUPRATM ”.

3.       row 162 - in image in row 158-159 the described encapsulation is not to seen

Response 3:

4.       row 165 - 10000 level image .. you will describe 100000 magnification, also row 166 form of the article:table 1 why doublespace , right column right hand sight

Response 4: Line 179-181, We have re-written this part according to the Reviewer’s suggestion. Line 111, Table 1 has made reasonable adjustments.

5.       121 - 10-9    -9 upper case

Response 5: Line 131,“10-9”were corrected as10-9.

6.       fig. 3 no mu marker clearly to seen, no information about magnification, working distance

Response 6: Line 186, We have added a description of the magnification. Line 179-181, we have made reasonable adjustments.

7.       fig. 4 the eds spectra are nothing readable, why energy range over 10 keV nothing to seen

Response 7: We adjusted the EDS information of Figure 4. Fe-Kα=6.3996 keV, C Kα=0.2774,S- Kα=2.3075,Ca- Kα=3.6905,Zn- Kα=8.6313,Al- Kα=1.4866,P- Kα=2.0134,Si- Kα=1.7398. All elements tested have Kα energy within 10 keV.

8.       fig 6 the legend is not readable

Response 8: We have made correction according to the Reviewer’s comments. We have added the font size in the Figure 6.

9.       fig 11 and 12 the text input is to small not readable

Response 9: fig 11 and 12 the text input has been enlarged.

10.    how measure the morphology in fig 13. The method is not clearly described AFM?, the x,y,  z-axes levels are not readable why conclusions in doublespace written?

Response 10: Special thanks to you for your good comments. We use 3D microscope with super wide depth of field to measure. Line 144-145, 333, We have made correction according to the Reviewer’s comments. Figure 13 does not indicate the x, y, and z axis information, probably because the size and high and low values of each graph can be clearly seen. Conclusions have made correction.

Thank you very much for consideration!

Sincerely Yours,

Chuanyi Xue

Round  2

Reviewer 1 Report

The authors have really improved the manuscript, however I still have some hesitations:

English should be revised because still some typos appear (like in line 158 "Carbon lubricants was", instead of "Carbon lubricants were").

The authors have included the viscosity values of the lubricants, but I am really surprised with these results. Carbon nanophases in small quantities have a great impact on the viscosity values of oil bases (Tribology International, 116, 371-382, 2017). However, the authors claim that with their measurements there is a small effect. I will feel more confident if they can show some bibliography regarding this matter.

Author Response

Dear Editors and Reviewers:

Thank you for your letter and for the reviewers’ comments concerning our manuscript entitled “Tribological performance of Nanocomposite Carbon lubricant additive” (ID: materials-403185). Those comments are all valuable and very helpful for revising and improving our paper, as well as the important guiding significance to our researches. We have studied comments carefully and have made correction which we hope meet with approval. Revised portion are marked in red in the paper. The main corrections in the paper and the responds to the reviewer’s comments are as flowing:

Point 1: Major points

The authors have really improved the manuscript, however I still have some hesitations:

 1.       English should be revised because still some typos appear (like in line 158 "Carbon lubricants was", instead of "Carbon lubricants were").

2.       The authors have included the viscosity values of the lubricants, but I am really surprised with these results. Carbon nanophases in small quantities have a great impact on the viscosity values of oil bases (Tribology International, 116, 371-382, 2017). However, the authors claim that with their measurements there is a small effect. I will feel more confident if they can show some bibliography regarding this matter.

Response 1: Line 158, "Carbon lubricants was" are corrected as" Carbon lubricants were ".

Response 2: Special thanks to you for your good comments. We are very sorry. The previous description is not very clear. The study of nanocomposite carbon is very interesting. Line 120-123, We have re-written this part according to the Reviewer’s suggestion. The content of nanocomposite carbon additive is 3%, which is not 3% of nanocomposite carbon. Then, Nanocomposite Carbon additive dispersion solution was prepared by mixing T161A high molecular weight polyisobutenyl succinimide and 350SN lubricants in a 1:1 ratio. The preparation of the nanocomposite carbon additive is a mixture of the modified nanocomposite carbon and the dispersion solution ratio of 1:39. The 3% ratio refers to the ratio of Nanocomposite Carbon additives to 350SN lubricants.Therefore, the true content of nanocomposite carbon is 0.075 wt.%.Since the detected kinematic viscosity is averaged over multiple measurements And, the actual change in kinematic viscosity is within the error range. So, the results of the kinematic viscosity in the article are credible.

 Thank you very much for consideration!

Sincerely Yours,

Chuanyi Xue

 Reviewer 2 Report

Line 130 Typo… K alpha

Line 181… Figure 3c corresponds to 6000, and not 100000 as noted in the text.

Figure 3… use “x1000” instead of “1000 times”… “c” corresponds to a lower magnification that “d”? There is some mistake there.

About experimental procedure. In Table 1… you show initial materials properties. That table shows that Ra of contact specimens are 0.43 and 0.45 microns. You perform the first test with a certain mixture carbon-lubricant and then you clean the parts for 15 minutes (as explained in lines162…). Then you use another lubricant and perform a new test using cleaned parts… and repeated the procedure. If this is the procedure used… you cannot compare your different mixtures as you are starting each experiment with a Ra different, as previous test had modified the surface and the roughness, as you effectively measured. If this is the procedure results are not reliable and the work can not be published. It the procedure was another, authors must clarify it.

Author Response

Dear Editors and Reviewers:

Thank you for your letter and for the reviewers’ comments concerning our manuscript entitled “Tribological performance of Nanocomposite Carbon lubricant additive” (ID: materials-403185). Those comments are all valuable and very helpful for revising and improving our paper, as well as the important guiding significance to our researches. We have studied comments carefully and have made correction which we hope meet with approval. Revised portion are marked in red in the paper. The main corrections in the paper and the responds to the reviewer’s comments are as flowing:

1.       Point 1: Line 130 Typo… K alpha

 2.       Line 181… Figure 3c corresponds to 6000, and not 100000 as noted in the text.

 3.       Figure 3… use “x1000” instead of “1000 times”… “c” corresponds to a lower magnification that “d”? There is some mistake there.

 4.       About experimental procedure. In Table 1… you show initial materials properties. That table shows that Ra of contact specimens are 0.43 and 0.45 microns. You perform the first test with a certain mixture carbon-lubricant and then you clean the parts for 15 minutes (as explained in lines162…). Then you use another lubricant and perform a new test using cleaned parts… and repeated the procedure. If this is the procedure used… you cannot compare your different mixtures as you are starting each experiment with a Ra different, as previous test had modified the surface and the roughness, as you effectively measured. If this is the procedure results are not reliable and the work can not be published. It the procedure was another, authors must clarify it.

Response 1: Line 132, the statements of “K a” were corrected as “ K α”                    

Response 2-3: Line 181-183, 188, We have made correction according to the Reviewer’s comments.

Response 4: Line 165-166, We are very sorry that there is no clear description. The friction and wear test is very interesting. The surface is polished before each test. The friction surface wear test is performed after the surface has reached the roughness standard (0.43 and 0.45 microns). Therefore, the test data is accurate.

 Thank you very much for consideration!

Sincerely Yours,

Chuanyi Xue

Reviewer 3 Report

row 140 The SEM Supra TM 55 is from Zeiss!

fig. 13: the letter size is to small, the informations are not readable!

Author Response

Dear Editors and Reviewers: Thank you for your letter and for the reviewers’ comments concerning our manuscript entitled “Tribological performance of Nanocomposite Carbon lubricant additive” (ID: materials-403185). Those comments are all valuable and very helpful for revising and improving our paper, as well as the important guiding significance to our researches. We have studied comments carefully and have made correction which we hope meet with approval. Revised portion are marked in red in the paper. The main corrections in the paper and the responds to the reviewer’s comments are as flowing:

Point 1:

1.       row 140 The SEM Supra TM 55 is from Zeiss!

2.       fig. 13: the letter size is to small, the informations are not readable!

        Response 1: Line 142, We have made correction according to the Reviewer’s comments.

        Response 2: Line 352, We have added the data in the figure. AndWe have enlarged Figure 13.

 Thank you very much for consideration!

Sincerely Yours,

Chuanyi Xue